gllvm 2.0: fast fitting of advanced ordination methods and joint species distribution models

Korhonen Pekka pekka.o.korhonen@jyu.fi 1
Hui Francis K.C. 2
Niku Jenni 3
Taskinen Sara 1
van der Veen Bert 4
1 Department of Mathematics and Statistics, University of Jyväskylä , Jyväskylä , Finland
2 Research School of Finance, Actuarial Studies and Statistics, Australian National University , Canberra , Australia
3 Faculty of Sport and Health Sciences, University of Jyväskylä , Jyväskylä , Finland
4 Department of Mathematical Sciences, Norwegian University of Science and Technology , Trondheim , Norway
Brygadyrenko Viktor
Electronic publication date: 2025 Dec 12
Publication date: 2025
Volume: 13
Electronic Location ID: e20338
Received 2025 May 6; Accepted 2025 Oct 13
Copyright: ©2025 Korhonen et al.
Copyright year: 2025
Copyright holder: Korhonen et al.
License: This is an open access article distributed under the terms of the Creative Commons Attribution License, which permits unrestricted use, distribution, reproduction and adaptation in any medium and for any purpose provided that it is properly attributed. For attribution, the original author(s), title, publication source (PeerJ) and either DOI or URL of the article must be cited.
License URL: https://creativecommons.org/licenses/by/4.0/

Keywords: Joint modeling, Maximum likelihood, Multivariate abundance data, Nested design, Ordination, Phylogenetic mixed models, R, Software

Funding: The Kone foundation The Research Council of Finland 356484 HiTEc COST Action CA21163 An Australian Research Council Discovery Project DP240100143 Pekka Korhonen, Jenni Niku and Sara Taskinen were funded by the Kone foundation. Sara Taskinen and Pekka Korhonen were funded by the Research Council of Finland (356484) and the HiTEc COST Action (CA21163). Francis K.C. Hui was funded by an Australian Research Council Discovery Project (DP240100143). There was no additional external funding received for this study. The funders had no role in study design, data collection and analysis, decision to publish, or preparation of the manuscript.

==============================
Background

Over the past decade, joint species distribution models (JSDMs) and model-based ordination have emerged as powerful tools for the analysis of community ecology data. Generalized linear latent variable models (GLLVMs) offer a flexible framework for multivariate analysis of a wide range of data types, based on including a small number of latent variables to perform dimension reduction while accounting for residual correlation between species.

Fast estimation methods

The R package gllvm implements a wide range of GLLVMs, with estimation performed via fast approximate likelihood-based techniques; including the recently proposed extended variational approximation, which is applicable to almost any combination of response type and link function. Since its original development and accompanying software paper, the gllvm package has undergone a significant overhaul, consolidating its place as a general framework for joint modeling of community ecology datasets.

Expanded functionalities

Some of the key new features of gllvm include model-based constrained and concurrent ordination methods, capacity to account for nested/hierarchical sampling designs, and (phylogenetic) random effects. On top of this, other notable improvements include a great expansion of the response types that it can handle, enhanced capabilities of GLLVM inference, selection and prediction, and an easier-to-use interface for model fitting.

Introduction

The last decade has seen a push for model-based multivariate methods. Such multivariate methods extend generalized linear models to multiple response variables, usually while incorporating correlation between the responses. The correlation structure is induced by latent variables, hence the models are now commonly referred to as “Generalized Linear Latent Variable Models,” or GLLVMs; although, originally, also the term “Generalized Linear Latent and Mixed Models” has been used (Skrondal & Rabe-Hesketh, 2004), serving to perhaps better highlight the place of the framework inside the wider statistical literature.

Although GLLVMs are potentially applicable to a wide range of biological fields, such as morphometrics or quantitative genetics, most of the developments have instead targeted community ecology; a field of study with a rich history of multivariate method development, such as for classical ordination methods in the '80s and '90s (Kenkel & Orlóci, 1986; ter Braak & Prentice, 1988; Philippi, Dixon & Taylor, 1998). In community ecology, the data consist of observations from an assemblage of potentially interacting species (or other taxonomic groupings), usually collected from a set of samples that are spatially and/or temporally structured. The data are often very sparse, because species often occur at few places due to, for example, environmental filtering (Ovaskainen et al., 2017). The potential to study the effects of biotic filtering acted as the catalyst for increasing popularity of GLLVMs in community ecology—as a technical solution for implementing joint species distribution models (JSDMs) (Pollock et al., 2014; Warton et al., 2015). GLLVMs make for a fast and technically efficient tool for implementing JSDMs, but it is in the application of ordination methods that the latent variable approach fully comes to fruition.

Classically, community ecological studies visualize a small number of latent variables—or, in ecological terms, environmental gradients—to describe patterns of species co-occurrence as well as the (dis)similarity of sites, using either indirect/unconstrained or direct/constrained ordination; for more on this distinction, see ‘Scottish ground beetle dataset’. In such a context, and for sparse data, GLLVMs are the perfect tool for multivariate analysis; by reducing the number of parameters in complex statistical models, the framework greatly facilitates community ecological studies. With a flexibility that is unparalleled by any other ordination method, GLLVMs offer a wide range of opportunities to account for the properties of ecological data by specification of a response distribution and potential inclusion of random effects. The models can straightforwardly be adjusted for good correspondence with the ecological processes under study. Above all, GLLVMs provide a single framework that covers an extremely wide class of models for multivariate analysis (Warton et al., 2015; Niku et al., 2021; Van der Veen et al., 2021; Van der Veen et al., 2023).

GLLVMs have been consistently shown to outperform traditional unconstrained ordination techniques such as principal component analysis and non-metric multidimensional scaling, as well as constrained ordination methods such as redundancy analysis or canonical correspondence analysis (Hui et al., 2015; Van der Veen et al., 2023; Korhonen et al., 2024); the many advantages including e.g., more flexibility in specifying the mean-variance relationship, quantification of uncertainty, and availability of tools for model comparison and diagnostics (Warton, Wright & Wang, 2012). In this article, we provide an overview of the functionality that the gllvm R package has accummulated over the last six years, for fast fitting of advanced ordinations and JSDMs.

The R package gllvm (Niku et al., 2025)—available from the Comprehensive R Archive Network (CRAN)—was originally conceived for fast fitting of multispecies models for community ecology, given that at the time of first development the R software landscape (R Core Team, 2016) was lacking; there were few packages that could feasibly fit JSDMs to large datasets with a non-negligible number of species. Its computational efficiency is due to the implementation of likelihood-based methods coupled with a variety of approximation approaches as developed in Niku et al. (2017b) and Hui et al. (2017). However, since its original development and accompanying software article (Niku et al., 2019b), the functionalities of the package have been greatly expanded, so much so that it can now be considered a general framework for joint modeling of community ecology data. A summary of many of the new features since Niku et al. (2019b) is presented in Table 1, but among these they include: capacity to handle a much wider range of response distributions (see Table A1 in Section ‘Response distributions available in gllvm’ in Appendix), advanced model-based constrained and concurrent ordination techniques, the possibility of accounting for correlations both within and between species due to nested/hierarchical sampling designs, phylogenetic random effects models, and greatly enhanced tools for estimation and statistical inference. The goal of this paper is to present this new, significantly overhauled gllvm package, using a number of worked examples to illustrate how the above new features allow ecologists to fit a myriad of multivariate models to community ecology data.

Table 1 Overview of existing and new features in gllvm, categorized by some of the main elements and analytical tasks the package (now) contains.

	Existing features	New features	
Model type	Linear, independent LVs; Fourth-corner GLLVM	Correlated LVs; Quadratic LVs; LVs informed by covariates; Random slopes for covariates; (Phylogenetic) random effects; Reduced-rank regression	
Response type	Continuous; Presence-absence; (Overdispersed) counts; Ordinal; Non-negative continuous	Zero-inflated counts; Positive continuous; Percent cover (with 0% and/or 100% records)	
Community-level row effects	Single fixed/random	Multiple fixed/random; Correlated/structured effects	
Ordination analysis	Unconstrained; Residual	Constrained; Concurrent; Partial constrained/concurrent	
Species associations	Residual correlation	Environmental correlation	
Inference	Analysis of deviance; Confidence intervals for parameters; Diagnostic residual plots	Fixed-effects covariance matrix; Prediction intervals; Variance partitioning; Capacity to handle missing (MAR) data	
Visualization	Ordination (bi-)plots; Plots of estimated fixed effects;	Uncertainty regions in ordination; Plots of predicted random effects Variance partitioning plot	
Model fitting methods	Laplace approximation; Gaussian variational approximation	Extended variational approximation; Parallel computation	

Broadly speaking, gllvm can now fit GLLVMs with one or more of the following three parts: (1) latent variables; (2) species effects; (3) community-level row or sample effects. Each of these components corresponds to a specific formula argument in the package, namely  lv.formula

,    formula

, and    row.eff

, respectively, and all capable of accepting varying forms of covariates, phylogenetic information, and space or time-related information in the form of distance matrices, among other arguments. Below, we present two thoroughly worked examples to highlight the critical details of each of the three components, although they are not exhaustive and are further supported by vignettes available at https://jenniniku.github.io/gllvm/index.html.

The remainder of this article is structured as follows. In Section ‘Core components of the package and Estimation’ we briefly review the core components of GLLVMs as well as methods used for fast model fitting, respectively. Section ‘Worked examples’ illustrates some of the key new functionalities of the gllvm package outlined in Table 1, and how they are executed through the aforementioned three formula arguments. Section ‘Discussion’ closes the paper with some discussion and future outlook. Note, that portions of the text have been published as part of a preprint in Korhonen (2025).

Core components of the package

Let yij denote the record for response (species) j = 1, …, m recorded at sample i = 1, …, n e.g., study sites. We may also have information in the form of k environmental or habitat variables for each sample, denoted here as xi = (xi1, …, xik)⊤, and q trait variables for each species, denoted here as tj = (tj1, …, tjq)⊤. Finally, phylogenetic information may be available on the genetic relationship between the species in question, which arises in the form of a m × m correlation matrix.

For joint species distribution modeling, GLLVMs regress the mean abundance μij = E(yij) of the sample-species record against xi and a small number d ≪ m of latent variables, ui = (ui1, …, uid)⊤. As such, (almost) all models available in the gllvm package can be formulated as (1) gμij=ηij=αi+β0j+xi⊤βj+ui⊤γj,

with the one exception being GLLVMs where species respond unimodally i.e., in a quadratic manner to the latent variables (Van der Veen et al., 2021). In Eq. (1), g(⋅) denotes a known link function e.g., logit/probit-link for presence-absence data and log-link for count data, β0j denote species-specific intercepts, and αi are optional sample- or community-level effects, which can further be treated via a mixed-effects model as αi=xi⊤β+zi⊤λ, where zi denotes the ith row of a design matrix corresponding to the random effects term. The inclusion of such a model may be as simple as the need to perform a sample-level total abundance standardization, i.e., to account for known differences in sampling intensity across samples (Hui et al., 2015; Warton et al., 2015), or be more sophisticated such as accounting for temporal or spatial correlation between samples. We will also show later how to specify models for αi with (multiple) structured community-level row effects to account for nested/hierarchical sampling schemes, with potentially covariates at different levels of the hierarchy.

Let βj = (βj1, …, βjk)⊤ and γj = (γj1, …, γjd)⊤ denote the full vectors of species-specific coefficients related to the covariates, and species-specific loadings related to the latent variables, respectively. Note the latent variables ui themselves can be thought of as unmeasured environmental variables, or as sample (site) scores in an ordination, capturing the main drivers of species abundances’ or community composition. In the original work of Warton et al. (2015) and Niku et al. (2017a), the latent variables were assumed to be independent across samples and standard normally distributed, ui ∼ Nd(0, I). However, this assumption can now be relaxed in gllvm by assuming a temporal or spatial correlation structure for the latent variables, or by hierarchically regressing the latent variables against the covariates xi; the latter is similar to a typical constrained ordination.

Finally, the species-specific responses to the covariates βj can be regressed against trait covariates tj in order to explain interspecific variation in environmental responses. This is better known in community ecology as fourth-corner modeling (Niku et al., 2021), and can be formulated as (2) βj=βe+Bettj+bj,

where vector βe = (βe1, …, βek)⊤ denotes the main species-common effects for the covariates, k × q matrix Bet denotes the environment-trait interaction matrix also known as the fourth-corner matrix, and species-specific random effects bj = (bj1, …, bjk)⊤ are included and assumed to follow a normal distribution, bj ∼ Nk(0, Σb). To account for species’ non-independence, Eq. (2) can be generalized to incorporate phylogenetic information (Van der Veen & O’Hara, 2024). We note that the fourth corner GLLVM presented here is an extension on the (non-JSDM) fourth corner models of Jamil & ter Braak (2013) and Brown et al. (2014).

Estimation

Fitting GLLVMs is in general a computationally burdensome task. In the literature, many applications have so far employed an Expectation Maximization (EM) algorithm (Sammel, Ryan & Legler, 1997; Hui et al., 2015) or Markov Chain Monte Carlo (MCMC) methods (Tikhonov et al., 2020b; Pichler & Hartig, 2021). Given the computational intensity of EM and MCMC methods, a more feasible approach is to use methods which approximate the marginal likelihood function in a closed form. We review such methods briefly below, while the more technically oriented reader is directed to Section ‘Further details about likelihood-based estimation in gllvm’ in Appendix for further details.

In the gllvm package, we have implemented a variety of approximation approaches, coupled with automated differentiation techniques (Template Model Builder, Kristensen et al., 2016), that allow efficient model fitting for a plethora of response types and link functions (see Table A1 in Section ‘Response distributions available in gllvm’ in Appendix). In particular, compared with Niku et al. (2019b), a new feature of gllvm is the capacity for nearly universal model fitting, courtesy of the extended variational approximation (EVA) method of Korhonen et al. (2023). EVA presents a solution to a well-known drawback of standard Gaussian variational approximations (VA, e.g., Ormerod & Wand, 2010; Ormerod & Wand, 2012), where a closed-form expression for the variational lower bound is only available for a limited number of response distributions and link functions. For instance, previously in gllvm if one were to fit a GLLVM to say, presence-absence data with logistic link, one had to rely on Laplace’s approximation (LA, Tierney & Kadane, 1986) as closed-form objective function for VA was not available. By contrast, in cases where both LA and a closed-form VA are available e.g., presence-absences using a probit link function, the former has been shown to be both faster and typically more accurate (e.g., Niku et al., 2019a; Korhonen et al., 2023; Korhonen, Nordhausen & Taskinen, 2024).

To overcome the above limitations, EVA applies an additional approximation on the variational objective function in the form of a second-order series expansion. Thus, EVA can be viewed as a blend of the variational and Laplace approximations. Indeed, in the context of machine learning (Wang & Blei, 2013) proposed a similar estimation approach, aptly named delta method variational inference. Critically, the extra approximation step yields closed-form objective functions for practically any response type or link function. At the same time—similar to VA—EVA has been shown to perform well on both simulated and real data, managing to compete and often outperform LA in terms of speed and estimation accuracy (Korhonen et al., 2023; Korhonen et al., 2024; Korhonen, Nordhausen & Taskinen, 2024). In short, EVA has greatly diversified the number of response types and link functions that are available under the variational framework in gllvm, among them the logistic models for presence-absence or percentage data, and most recently the ordered beta distribution of Kubinec (2023) for fitting GLLVMs to sparse percent cover data models. Technical details regarding EVA can be found in Section ‘Extended variational approximations’ in Appendix.

Worked examples

In this section, we provide practical demonstrations using sample code for the overhauled gllvm package, focusing particularly on how to utilize the formula arguments corresponding to the three components of GLLVMs, different types of ordination analysis available, estimation using EVA, and some of the newly supported response types. In Section ‘Scottish ground beetle dataset’, we consider model-based constrained and concurrent ordination in order to extract reduced-rank representations of the environment for zero-inflated overdispersed count data involving lots of covariates. Next, in Section ‘Californian kelp forest data’, we present examples involved with fitting GLLVMs to percent cover data incorporating a nested sampling design, utilizing structured and correlated community-level row effects and latent variables, and leveraging phylogenetic information via a random effects formulation.

The example analyses were conducted in R (v4.4.0, R Core Team, 2016), using the gllvm release v2.0.4, available from https://github.com/JenniNiku/gllvm/releases or https://zenodo.org/records/15720641. Note, that there can exist slight deviations in results depending on the package or R version used. In general, the latest GitHub build can be installed e.g., with devtools (Wickham et al., 2022) by:

 > devtools::install_github("JenniNiku/gllvm")

From CRAN, the package is available simply with     install.packages("gllvm")

.

Scottish ground beetle dataset

The ground beetle dataset of Ribera et al. (2001) consists of measured counts from m = 68 species of ground beetles collected on n = 87 study sites across the Scottish landscape. Notably, the data contain k = 17 primary environmental covariates, among them e.g.: organic content, soil pH, moisture, canopy height, stem density, flower and fruit biomass, elevation in meters above sea level (m.a.s.l.), and management index score. While it would be possible to fit a multivariate generalized linear model (GLM, Wang et al., 2012) to such data, a useful alternative when the number of covariates is non-negligible and/or when a low-dimensional visual presentation of the species-environment is preferred to investigate a reduced-rank representation of the covariate parameter space (Yee & Hastie, 2003). One may also be interested in ascertaining which environmental covariates drive the gradients of this low-dimensional parameter space, and hence govern community composition (ter Braak & Prentice, 1988; ter Braak & Šmilauer, 2015). The data is included in gllvm and can be loaded with    data("beetle")

.

To address such questions, the gllvm package allows regression models to be formulated for the latent variables and associated species-specific loadings components, also referred to as model-based constrained and concurrent ordination respectively, after Van der Veen et al. (2023). This section provides worked examples to illustrate these new methods, together with the recently added zero-inflated negative binomial response distribution; see also https://jenniniku.github.io/gllvm/articles/vignette6.html for further examples of such ordinations.

Reduced-rank regression and constrained ordination

Consider first the case of using a standard multivariate GLM with only species-specific intercepts β0j, and species-specific coefficients βj. With the k = 17 covariates X in the ground beetle dataset, then the resulting model (3) ηij=β0j+xi⊤βj,

involves 68⋅(17 + 1) = 1,224 regression coefficients. Given the large number of parameters relative to the information available in the data, a reduced-rank regression (RRR) model (i.e., constrained ordination) significantly reduces the number of parameters to estimate, alleviates potential overfitting issues, and additionally offers the opportunity to construct a lower dimensional visualization (i.e., an ordination). In the constrained ordination, we introduce a k × d matrix of reduced-rank or canonical coefficients B, along with species-specific loading vectors denoted by γj, and impose the following structure on the regression coefficients βj in Eq. (3): (4) βj≈dBγj.

Alternatively, the constrained ordination model resulting from substituting Eqs. (4) into (3) can be written as a special case of the GLLVM in Eq. (1), by defining d-vector latent variables of the form ui = B⊤xi. Note that the identifiability constraints needed in order to fit such a model are quite different from a standard GLLVM where the ui are random effects; see Van der Veen et al. (2023) for technical details. Regardless, the latent variables (or site scores) ui are then said to be constrained by the environmental covariates xi i.e., so that we target only the part of the covariation in the species records that can be filtered by the covariates. In gllvm, such a constrained GLLVM can be fitted as follows:

> X <- scale(beetleEnv)    # scale  and  center  the  environmental  covariates > ftConstOrd  <- gllvm(y=beetle, X=X, family="negative.binomial", num.RR=2).

The familiar     summary

() function can then be used to inspect the resulting fit, while point estimates and corresponding 95% confidence intervals for the the canonical coefficients B can be visualized in the form of a caterpillar plot (Fig. 1 bottom right) via the command    plot(summary(ftConstOrd))

.

Figure 1 Top row: coefficient plots of the reduced-rank approximated species-specific covariate effects corresponding to reproductive biomass (left) and elevation in m.a.s.l. (right), for 34 of the total of 68 species.

Bottom left: model-based constrained ordination of the site scores; longer arrows indicate covariates with the largest relative effect, while arrows for which the 95% confidence interval of the associated slope in B excludes zero, are shown in a darker red color. Bottom right: point estimates and corresponding 95% confidence intervals for the canonical coefficients B. The plots are obtained based on fitting a negative binomial GLLVM with d = 2 constrained latent variables and k = 17 covariates.

In the code snippet above, the argument    num.RR

controls the number of constrained latent variables, noting this may be chosen using data-driven approaches such as information criteria or regularization penalties (e.g., Hui, Tanaka & Warton, 2018). Also, the canonical coefficient matrix B can be specified as random effects, with three options available for the associated variance components as available via an associated    randomB

argument; (i)    randomB="LV"

assumes the variance components to be specific to each latent variable. This induces a covariance structure on how the species’ responses are shaped by the covariates; (ii)    randomB="P"

introduces additional covariate-specific variances. The latter can serve to act as a type of shrinkage by encouraging some of the covariate effects to be shrunk towards zero. The    lv.formula

argument can alternatively be specified as a random effects formula with    randomB="P"

to incorporate correlation parameters between the covariate effects; (iii)    randomB="single"

assumes shared variance components across both LVs and covariates. Though by default the canonical coefficients are treated as fixed effects, treating the canonical coefficients as random-effects is usually preferred, particularly to stabilize estimation in the presence of strong multicollinearity between the elements of xi (Cajo ter Braak, pers. comm., 2022).

After fitting the GLLVM, constrained ordination plots (Fig. 1 lower left) along with coefficient plots for species-specific loadings (Fig. 1 top row) can be constructed as follows:

> ordiplot(ftConstOrd, symbols=TRUE, jitter=TRUE) > coefplot(ftConstOrd, which.Xcoef=c("Reprobiom", "Elevation"), ind.spp=c(1:34))

where the argument     jitter

was used to move the symbols around on the plot slightly to reduce visual cluttering, and    ind.spp

is used to select a smaller subset of species for which the coefficient estimates are plotted.

Concurrent ordination

In the constrained GLLVM Eq. (4), it is assumed that the latent variables are driven solely by the measured covariates. This is often unrealistic in practice, and so to simultaneously account for both measured and unmeasured drivers of species covariation we can instead fit a concurrent ordination GLLVM. That is, we can perform simultaneous constrained and unconstrained ordination by incorporating an additional set of “residual” latent variables, (5) ηij=αi+β0j+xi⊤Bγj+ϵi⊤γj,

where ϵi ∼ N(0, Σ) and Σ = diag(σ2) is a diagonal d × d matrix with variances σ2 as the diagonal elements. Critically, note the same vector of loadings, γj, is related to both rank-d terms: this allows us to alternatively formulate a regression model for the latent variables as ui = B⊤xi + ϵi. To contrast between unconstrained ui’s in Eq. (1), and the fully constrained latent variables in Eq. (4), we refer to the ui’s in Eq. (5) as informed latent variables, meaning they are influenced but not fully constrained by the measured covariates. In the gllvm package, a concurrent ordination GLLVM can be fitted by using the argument    num.lv.c

which controls the desired amount of informed LVs:

> ftConcOrd  <- gllvm(y=beetle, X=X, family="ZINB", num.lv.c=2, n.init=5)

Afterward, the ordination can be visualized with a call to     ordiplot()

, where—as with other types of ordination in gllvm—the argument    biplot=TRUE

can be used for specifying the resulting plot (see Fig. 2) to also include the estimates for the species’ loadings γj.

Figure 2 Model-based concurrent ordination of the ground beetle dataset, based on fitting a zero-inflated negative binomial GLLVM with d = 2 informed latent variables and k = 17 covariates.

Longer arrows indicate covariates with larger relative effects, while arrows for which the 95% confidence interval of the associated element in B excludes zero in both dimensions are shown in a darker red color—here meaning only the coefficients for canopy height. Estimates for the loadings γj are illustrated using the species’ labels in blue. Note, that the argument ind.spp was used to only plot the loadings for 34 of the species.

Being a more complex type of model, concurrent ordination may sometimes benefit from the use of the argument    n.init

(as above) in order to fit properly. In particular,    n.init=5

constructing starting values based on five initial quick runs of the GLLVM. The present functionality also involves taking into account whether convergence actually improves, by comparing the magnitudes of the gradient; this is generally done to increase the chances of reaching a stable maximum likelihood estimate. Finally, in some scenarios, it may be beneficial to start the estimation procedure from zero with the argument    starting.val="zero"

, or to try swapping between different optimization methods via the setting    optimizer

.

As an aside, the assumption that the two terms in Eq. (5) share the same species-specific loadings γj, can be relaxed in gllvm by combining constrained and unconstrained ordination terms via joint usage of the arguments    num.RR

and    num.lv

; this results in a so-called hybrid ordination. In such cases, the two ordinations can be visualized separately, though arrows can be drawn only for the constrained ordination part of the GLLVM, because the unconstrained ordination need not be orthogonal to the covariate effects as long as the number of constrained latent variables is smaller than the number of covariates. Equivalently, we can view such a model as a kind of JSDM with reduced-rank species-specific effects for the covariates, combined with a latent variable model to account for residual species associations.

Partial concurrent or constrained ordination

In some scenarios, one may wish to include full-rank effects for a subset of the environmental covariates while using a low-rank presentation for the remaining set. This may be thought of as “conditioning” the ordination on some of the measured xi’s, effectively removing their effect from the ordination. Such a “partial ordination” can be constructed in gllvm as in the following example: suppose we wanted to employ full-rank effects for canopy height and reproductive biomass, but a reduced-rank structure for the remaining 15 covariates. Then we can achieve this via the    formula

arguments:

> ftPartOrd  <- gllvm(y=beetle, X=X, family="ZINB", num.lv.c=2,                formula=~Canopyheight + Reprobiom,                lv.formula=~Texture + Org + pH + AvailP + AvailK + Moist                              + Bare + Litter + Bryophyte + Plants.m2                              + Stemdensity + Biom_l5 + Biom_m5 + Elevation                              + Management, randomB="P", n.init=5).

In the above, we used     randomB="P"

so that the canonical coefficients B are assumed to be random effects with covariate-specific variance components. Alternatively, we could incorporate correlation parameters between the reduced-rank covariate effects by specifying a random effects formula as follows:

lv.formula=~(0 + Texture + Org + pH + AvailP + AvailK + Moist                       + Bare + Litter + Bryophyte + Plants.m2 + Stemdensity                       + Biom_l5 + Biom_m5 + Elevation + Management|1).

Furthermore, the full-rank effects can be treated as random slopes (e.g., Nussey, Wilson & Brommer, 2007) by instead using     formula=~(0 + Canopyheight|1) + (0 + Reprobiom|1)

, for independent effects, or with    formula=~(0 + Canopyheight + Reprobiom|1)

, for correlated effects. As noted in Section ‘Reduced-rank regression and constrained ordination’, both these options encourage shrinkage e.g., from the resulting output presented below we can see that relative to earlier models with fixed effects, the predicted effects of some of the covariates have been shrunk very close to zero:

> round(cbind(ftPartOrd$params$LvXcoef, ftConcOrd$params$LvXcoef[-c(11,15),]),  digits=5)                    CLV1      CLV2      CLV1      CLV2 Texture        0.01627   0.02919   0.35606   0.38039 Org           -0.04452   0.17585  -0.82587   0.35149 pH            -0.00795   0.64358  -0.15316   0.79692 AvailP        -0.01660  -0.32597  -0.22125  -0.35552 AvailK        -0.00001   0.00000  -0.00914   0.07170 Moist          0.06890  -0.24147   1.15432  -0.14245 Bare          -0.02380  -0.48900  -0.50632  -0.96130 Litter        -0.01490  -0.15635  -0.29376  -0.29661 Bryophyte     0.03446  -0.02939   0.68243   0.19831 Plants.m2    -0.01499   0.02439  -0.26844   0.05678 Stemdensity   0.00834  -0.00030   0.15938  -0.07107 Biom_l5      -0.03822  -0.59255  -0.85311  -1.02737 Biom_m5        0.00003  -0.00001   0.31475  -0.13100 Elevation     0.00892  -0.48093   0.20535  -0.53644 Management   -0.06214   1.17037  -1.01267   1.06269.

In Section ‘Species correlations due to random covariate effects in Appendix’, we also demonstrate how to use the function     getEnvironCor()

together with the package corrplot, to plot species correlations that are due to environmental covariate random effects, based on (partial) constrained or concurrent ordination models.

To summarize, gllvm now permits a wide variety of GLLVMs based on “mixing-and-matching” various latent variable configuration of the    formula

and    lv.formula

formula arguments, together with the    num.lv

(unconstrained),    num.RR

(constrained), and    num.lv.c

(informed) options. Beyond this, the argument    quadratic

can be also used to incorporate unimodal responses of species to (virtually) any of the GLLVMs above. Effectively, this takes the form, ηij=αi+β0j+ui⊤γj−ui⊤Djui,

where Dj denotes a positive-definite diagonal matrix of quadratic coefficients to the latent variables ui. We refer to Van der Veen et al. (2021) and https://jenniniku.github.io/gllvm/articles/vignette5.html for more information on such models and ordinations.

Zero-inflation and model selection

The model-based ordinations produced in Section ‘Concurrent ordination’ and Section ‘Partial concurrent or constrained ordination’ assumed zero-inflated count distributions for the ground beetle species records (Lambert, 1992; Greene, 1994). In particular, the zero-inflated Poisson and zero-inflated negative binomial distributions are now available in gllvm, and are suited to situations where a standard count model is not capable of explaining the observed rate of zero records for one or more species. Briefly, these distributions are defined by the probability mass function: PYij=0=πj+1−πj⋅pYij=0,PYij=c=1−πj⋅pYij=c,c=1,2,…,

where p(⋅) denotes the assumed distribution for the (potentially overdispersed) count process i.e., a Poisson or negative binomial distribution, and πj are species-specific parameters controlling the level of zero-inflation for species j = 1, …, m. The choice between the zero-inflated Poisson versus zero-inflated negative binomial models is similar to the choice of the standard Poisson versus negative binomial distributions. That is, the latter (already) accommodates overdispersion by including additional species-specific dispersion parameters ϕj (these can also be shared; see the examples in the following section). We refer to Feng (2021) and references therein for general discussion on zero-inflated models.

Finally, while for model-based ordination it is natural to employ d = 2 to 3 latent variables for the purposes of visualization, gllvm now offers some data-driven procedures for selecting on the final model, whether this is the choice of d, the covariates to retain in various parts of the GLLVM, the usage of zero-inflated versus standard counts distribution, and among other decisions. Specifically, the Akaike, corrected Akaike, and Bayesian information criteria (e.g., Burnham & Anderson, 2002) are available by calling either    AIC()

,    AICc()

, or    BIC()

, respectively, on a fitted gllvm object. Table 2 contains the value of the three information criteria for (zero-inflated) Poisson and negative binomial GLLVMs fitted to the ground beetle dataset, while also varying the number of informed latent variables (    num.lv.c

) from two to four. All three criteria select the negative binomial GLLVM, although with differing numbers of informed latent variables.

Table 2 Information criteria (Akaike or AIC, corrected Akaike or AICc, and Bayesian or BIC) values with the maximized log-likelihood value and degrees of freedom (the number of freely estimated parameters) for various concurrent ordination GLLVMs fitted to the ground beetle dataset, assuming different response distributions (family)  and varying the number of informed latent variables (num.lv.C) .

The model with the lowest value of each criterion is bolded.

family	num.lv.c	AIC	AICc	BIC	logL	df	
”Poisson”	2	107,706.96	107,726.66	109,284.72	−53,617.480	236	
	3	75,212.17	75,248.18	77,331.44	−37,289.083	317	
	4	53,562.34	53,619.31	56,209.76	−26,385.168	396	
”ZIP”	2	69,545.43	69,578.48	71,577.80	−34,468.716	304	
	3	49,521.64	49,575.39	52,095.53	−24,375.822	385	
	4	38,601.79	38,680.95	41,703.82	−18,836.895	464	
”negative.binomial”	2	18336.45	18,369.50	20,368.82	−8,864.225	304	
	3	18,107.78	18,161.52	20,681.66	−8,668.888	385	
	4	17,959.43	18,038.59	21,061.46	−8,515.713	464	
”ZINB”	2	18,438.87	18,488.94	20,925.85	−8,847.436	372	
	3	18,214.12	18,289.42	21242.61	−8,654.058	453	
	4	18,030.58	18,135.93	21,587.22	−8,483.288	532	

In addition to the information criteria, it is advisable to visually inspect the residuals after fitting the model. In gllvm this can be done simply by calling the function    plot()

on the model object. Specifically, the package utilizes Dunn-Smyth residuals (or randomized quantile residuals, Dunn & Smyth, 1996); resulting in continuous (normal) residuals, even for discrete response types. The residual plots are then assessed as akin to the case of a standard linear model. For more details on diagnostics, refer to Niku et al. (2019b).

Californian kelp forest data

In the second worked example, we consider a kelp forest dataset from the Santa Barbara Coastal Long Term Ecological Research site (SBC LTER, Reed & Miller, 2023), comprising measurements of percent cover of m = 130 species of marine macroalgae and sessile invertebrates collected between years 2000–2020 along a total of 44 permanent transect lines nested inside 11 observational sites. The data are very sparse, with 88% of the records being zeroes, while the largest recorded cover is 97%. Also, some of the sites were located on islands, while others were located across the coast. The dataset can be accessed within gllvm with the command    data("kelpforest")

. The data comes with two measured covariates, namely the rockiness of the seabed (%) at each location and the average number of giant kelp fronds. The latter was log-transformed, and afterwards both were scaled and utilized in all of the examples below. In addition, the dataset includes taxonomic information for most of the species encountered.

In contrast to the first example, we can see that kelp forest data involves a nested sampling design, which needs to be accounted for in the process of building a JSDM. More broadly, it is generally assumed that species records collected closer to each other in time and/or space will be more similar to each other. Analogously, species which are closely related in terms of their evolutionary history may respond similarly to a given environment. Through a worked example then, the kelp forest dataset allows us to demonstrate a number of newer features present in gllvm including but not limited to structured models for community-level row effects and latent variables, phylogenetic random effects, and response families for sparse percent cover data.

Structured and correlated community-level row effects

To account for transects nested within sites and years as part of the SBC LTER study, we can fit a GLLVM containing community-level row effects for sampling year, site and transect ID i.e., αi = αyear(i) + αsite(i) + αtran(i) in Eq. (1). In gllvm, accounting for such a nested design is possible via the argument    row.eff

, which now accepts formulas with fixed and/or random effects as exemplified below:

> ftStrucRow  <- gllvm(y=Ysess, X=Xenv, num.lv=0, family="orderedBeta",                         formula=~logKELP_FRONDSsc + PERCENT_ROCKYsc,                         studyDesign=Xenv[,c("SITE","TRANSECT","YEAR")],                         row.eff=~(1|SITE/TRANSECT) + YEAR,                         method="EVA", link="logit", disp.formula=shapeForm,                         setMap=setMap, zetacutoff=c(0,20)).

In addition to a fixed effect per sampling year, the syntax     row.eff=~(1|SITE/TRANSECT) + YEAR

specifies a random intercept for each of the 11 labeled study sites, along with a random intercept for transects nested within sites (noting there are total of 44 transects in the study). As community-level row effects, these are assumed to be the same across all species, accounting for differences in total percent cover across sites and transects. The indicator variables mapping the random intercepts to the site and transect ID are supplied as part of the    studyDesign

argument. Finally,    disp.formula

,    zetacutoff

, and    setMap

are related to the use of an ordered beta response distribution (    family="orderedBeta"

) coupled with a logit link for the species records (see Section ‘Models for sparse percent cover data’ later on for more details), while    method="EVA"

specifies the estimation to be done using the extended variational approximations as outlined in Section ‘Estimation’.

To further account for sampling variability, one may instead wish to replace the fixed effect of year with a random effect, where species compositions expressed by a given habitat are more similar to each other for years closer together e.g., the community-level row effects exhibit some sort of autoregressive correlation. Such a temporal correlation structure can be imposed using    row.eff=~(1|SITE/TRANSECT) + corAR1(1|YEAR)

as the row effect formula, where    corAR1

specifies a first-order autocorrelation structure on the random intercept for year. Other choices of correlation structures in gllvm include compound symmetry (    corCS

), exponentially decaying (    corExp

), and Matérn (    corMatern

) structures. The latter two choices are particularly common in the context of spatial GLLVMs (e.g., Ovaskainen et al., 2016; Tikhonov et al., 2020a), noting they require the user to also supply a matrix of coordinates via the argument    dist

.

Structured and correlated latent variables

Similarly to community-level row effects, the latent variables can also be structured and/or correlated, reflecting, e.g., presumed similarities regarding the unobserved environmental factors driving community composition, between observations in proximity in terms of time or space. For example, if    num.lv=2

and    lvCor=corAR1(1|YEAR)

is specified in our application to the kelp forest data, then each of the two latent variables will follow an autoregressive process, but with separate autocorrelation coefficients. That is, in Eq. (1) instead of ui, we define LVs as uyear(i) such that year(i) = t if sampling unit (row) i is from year t, with values in {0, 1, …, 20} (2000–2020). By defining uk. = (uk0, uk1, …, uk20)⊤, for k = 1, 2, the autoregressive structure is then formulated as uk. ∼ N(0, Σk), where the covariance matrices assume the form: Σk=1ρkρk2⋯ρk20ρk1ρk⋯ρk19⋮⋮⋮⋱⋮ρk20ρk19ρk18⋯1;ρk∈−1,1,k=1,2.

The code to fit such a model in gllvm to the kelp forest community data is given below:

> ftCorrLV  <- gllvm(Ysess, Xenv, num.lv=2, family="orderedBeta",                      formula=~logKELP_FRONDSsc + PERCENT_ROCKYsc,                      studyDesign=Xenv[,c("SITE","TRANSECT","YEAR")],                      row.eff=~SITE/TRANSECT, method="EVA", link="logit",                      disp.formula=shapeForm, lvCor=~corAR1(1|YEAR),                      zetacutoff=c(0,20),  starting.val="zero", setMap=setMap) > ftCorrLV$params$rho.lv   # estimates  for  the AR(1)  coefficients    rho.lv1    rho.lv2 0.9134419  0.9433707.

As can be seen, the estimated autoregressive terms (     rho.lv1

 and    rho.lv2

) are both relatively close to one, indicating strong temporal correlations between years. Similar to the case of the community-level row effects, gllvm permits a number of correlation structures, with the coordinate matrix (if required) supplied as part of the argument    distLV

. With GLLVMs involving multiple covariates along with structured/correlated community-level row effects and latent variables, a common question ecologists are interested in answering relates to quantifying the relative contributions of the different components in the model to the overall species covariation (see e.g., Ovaskainen et al., 2016; Bjork et al., 2018). Such variance partitioning can be performed straightforwardly in gllvm via the functions    varPartitioning()

and the associated plotting function    plotVP()

,

> varPart  <- varPartitioning(ftCorrLV, groupnames=c("Giant  kelp  frond                    density (log)", "Seabed  rockiness  (%)", "LV1", "LV2",                    "Site-transect  fixed  effect")) # more  descriptive  naming > plotVP(varPart, xlab="Species", las=2, cex.names=0.7, # plot() also  works                args.legend=list(cex=0.9),  col=hcl.colors(5,"TealRose")).

The resulting variance partitioning plot can be seen in Fig. 3. On average, the latent variables specific to sampling year explain ∼42% of the total covariation, the (log) number of giant kelp fronds contribute a relatively small amount except for a few species, while seabed rockiness and site-transect fixed effects each account for approximately 23% of the total species covariation.

Figure 3 Variance partitioning based on fitting an ordered beta GLLVM with d = 2 correlated latent variables (year), fixed structured community-level row effect specific to each site-transect pair, and k = 2 environmental covariates, to the kelp forest data.

Taken together, the two latent variables explain around two fifths of the total species covariation, although the relative contributions of the measured covariates and community-level row effects vary across species.

Phylogenetic random effect model

Another feature of the kelp forest data is that it contains taxonomic classifications, allowing us to construct a phylogenetic tree for the species. Following Van der Veen & O’Hara (2024), we demonstrate how phylogenetic random effect GLLMs can be fitted in gllvm. Structurally, such a model builds upon the fourth-corner formulation in Eq. (2) as follows: focusing on the full kelp forest data, consider dummy variables tj which indicate whether species j belongs to the group of sessile invertebrates (tj = 1) or macroalgae (tj = 0).

In a phylogenetic GLLVM then, for covariate l = 1, …, k and given a phylogenetic covariance matrix C, we have (6) b1l,…,bml⊤∼Nm0,σl2Cρl+1−ρlIm,

where ρl ∈ [0, 1] denotes the phylogenetic signal parameter for the l-th covariate, and noting gllvm allows this to be shared i.e., ρl = ρ for all l = 1, …, k covariates. If the signal ρl is estimated to be close to zero (one), then there is less (more) evidence that species responses are phylogenetically structured. Phylogenetic GLLVMs can be particularly useful when dealing with very sparse data, as rare species can borrow strength from more prevalent species which are closely (evolutionary) related in order to improve their estimation and prediction performance (Ovaskainen et al., 2017; Matsuba et al., 2024). Indeed, recall the kelp forest data is very sparse with over 80% of the species records being exactly zeros, making it a prime use case.

To fit a phylogenetic GLLVM using gllvm, we first have to build the phylogenetic tree and the covariance and distance matrices based on the taxonomic classifications of the species in the data. These are referred to as     tree

,    colMat

, and    dist

, respectively, in the code snippet below, and we provide details on defining these objects in Section ‘Phylogenetic tree and covariance matrix’ in Appendix. Next, the gllvm package makes use of a number of computational techniques to speed up fitting of this model, the details of which can be found in Van der Veen & O’Hara (2024). In particular, the estimation process greatly benefits from a new option to perform parallel computation now: after loading gllvm, we enable this using

> TMB::openmp(n=parallel::detectCores()-1,  autopar=TRUE)

where for illustration, we have used the maximum number of threads available minus one.

Suppose we order the species in the kelp forest data according to their distance from the root of the phylogenetic tree (stored in the variable     order

in the code below), although this is not necessarily the most optimal choice (see the associated vignette https://jenniniku.github.io/gllvm/articles/vignette7.html for diagnostic checks around the choice of ordering). Then along with an additional tuning parameter on the number of neighbors, which we leave here at the default value of    nn.colMat=10

 (see Section ‘Phylogenetic tree and covariance matrix’ in Appendix for details), the phylogenetic GLLVM and associated summary output is given as follows:

> ftPhylo  <- gllvm(y=Yphyl[,order], X=Xenv, TR=Trphyl[order,, drop=FALSE],                    formula=~(logKELP_FRONDSsc + PERCENT_ROCKYsc) +                               (logKELP_FRONDSsc + PERCENT_ROCKYsc): (GROUP),                    randomX=~logKELP_FRONDSsc + PERCENT_ROCKYsc, colMat=list(colMat[order,order], dist=dist[order,order]),                    colMat.rho.struct="term", nn.colMat=10,  beta0com=TRUE,                    method="EVA", link="logit", family="orderedBeta",                    n.init=3, disp.formula=shapeForm, zetacutoff=c(0,20),                    setMap=setMap, optim.method="L-BFGS-B", num.lv=0) > summary(ftPhylo) AIC:   11969.85  AICc:   11970.86  BIC:   13781.04  LL:   -5789 df:   196 Informed  LVs:   0 Constrained  LVs:   0 Unconstrained  LVs:   0 Formula:   ~logKELP_FRONDSsc+PERCENT_ROCKYsc+logKELP_FRONDSsc:GROUPINVERT                +PERCENT_ROCKYsc:GROUPINVERT LV  formula:   ~ 0 Row  effect:   ~ 1 Random  effects:  Name                Signal  Variance  Std.Dev  Corr  logKELP_FRONDSsc  0.0032  0.0395    0.1989  PERCENT_ROCKYsc   0.1073  0.0542    0.2327   -0.4436 Coefficients  predictors:                                    Estimate  Std. Error z value  Pr(>|z|) logKELP_FRONDSsc                0.021611    0.034766    0.622     0.534 PERCENT_ROCKYsc                 0.052367    0.059213    0.884     0.376 logKELP_FRONDSsc:GROUPINVERT  0.005584    0.047398    0.118     0.906 PERCENT_ROCKYsc:GROUPINVERT   0.112491    0.080530    1.397     0.162.

As seen above, there is little evidence of phylogenetic signals in this particular data, with the estimated values of ρ ˆ1=0.0032 and ρ ˆ2=0.1073. The associated community effects from the estimated phylogenetic GLLVM together with the associated tree is visualized in Fig. 4, obtained via the command     phyloplot(ftPhylo,tree)

.

Figure 4 A phylogenetic tree for species in the kelp forest data, together with the predicted species-specific random effects, based on fitting an ordered beta phylogenetic GLLVM model with two covariates and one functional trait.

We conclude this part of the worked example with a few remarks. First, as    num.lv

defaults to two latent variables unless we already include constrained (Section ‘Reduced-rank regression and constrained ordination’) or informed latent variables (Section ‘Concurrent ordination’), then we need to set    num.lv=0

in order to fit the model without any LV terms. Second, setting    colMat.rho.struc="term"

specifies covariate-specific phylogenetic signal parameters ρl, while a shared phylogenetic signal parameter ρ across the measured covariates could be specified instead via the argument    colMat.rho.struct="single"

. Finally, a limited memory optimization scheme is used via    optim.method="L-BFGS-B"

, which in our own experience usually works better for phylogenetic GLLVMs, while the argument    beta0com=TRUE

specifies one fixed intercept term shared by all species.

Models for sparse percent cover data

In the worked examples to the kelp forest data throughout Section ‘Structured and correlated community-level row effects’ to Section ‘Phylogenetic random effect model’, we fitted GLLVMs assuming an ordered beta distribution (Kubinec, 2023) for the species records yij. This was used to accommodate the sparse, percent cover nature of the responses i.e., continuous data that can take any value between and including zero and one, with a large percentage (over 80%) of zeros. The ordered beta GLLVM, together with the beta and hurdle beta hurdle GLLVMs, were introduced in the gllvm package through the recent work of Korhonen et al. (2024), and in doing so addressed an important gap regarding the how to fit JSDMs for multivariate sparse percent cover data. In this section, we briefly review these models and their availability within the package.

In ecology, percent cover data e.g., covers of sessile organisms as in the kelp forest data shown above, plant cover data (Elo et al., 2016; Elo et al., 2024), are typically sparse with a large percentage of zero records. This presents a problem for the standard beta GLLVM i.e., a latent variable model that assumes the responses come from the beta distribution(available in gllvm via    family="beta"

) as it cannot be applied directly to data containing exact zero or one records. While the addition of a small value to shift the responses away from exactly zero and one presents a popular ad-hoc solution, this is often not appropriate when the percentage of zero and one records becomes non-negligible; see also the work of OHara & Kotze (2010) on loosely related work advocating against the use of log transformations for count data.

As a more systematic alternative for multivariate sparse percent cover data, gllvm fits GLLVMs assuming a hurdle beta distribution for the responses by setting    family="betaH"

. Similar to zero-inflated models discussed in ‘Zero-inflation and model selection’ for count data, hurdle models introduce an external process to generate exact 0% (100%), with the principal difference bring that in a hurdle model this external process is assumed to be the sole source of the 0% and 100% records. In gllvm, the beta hurdle GLLVM is implemented as a two-part (namely the hurdle and zero-truncated count parts) model for percent cover data that includes exact zero records (although a three-part GLLVM that allowed for exact 100% records was also proposed in Korhonen et al., 2024), where the two parts share the same environmental covariates xi and latent variable scores ui, but separate regression coefficients {β0j, βj} and loadings γj. Conditional on presence, i.e., yij > 0, the response is modeled using a beta GLLVM via a logit link, while the presence-absence part is modeled by a logistic GLLVM.

Yet another alternative available in the gllvm package is the ordered beta distribution, as available via the argument    family="orderedBeta"

and used in the worked examples above. This model can be seen as being in-between the beta and hurdle beta GLLVMs in terms of model complexity, being able to accommodate exact zero and one records but using fewer parameters than a hurdle beta GLLVM by coupling the distinct processes together under one linear predictor. That is, the ordered beta GLLVM uses same linear predictor ηij for all parts of the distribution, but incorporates two ordered cutoff parameters reminiscent of a logistic proportional odds model (McCullagh, 1980) to separate between the three different classes the response can take, i.e., {0}, (0, 1) and {1}. The current implementation of ordered beta GLLVMs in the package allows for both cutoff parameters to be shared across all species (    zeta.struc="common"

), or be species-specific lower cutoff but species-common upper cutoff parameters (    zeta.struc="species"

). The reasoning behind the latter is that for most multivariate sparse percent cover datasets, we typically observe exactly 100% cover for only a handful of records, compared to exact 0% cover typically being recorded at least once for nearly all species e.g., in the kelp forest data 88% of the total records are zeros, while the biggest recorded cover is that of 97%. Conditional on yij ∈ (0, 1), the response is again modeled using a beta GLLVM via a logit link; see Kubinec (2023) and Korhonen et al. (2024) for more details on the formulation.

Finally, as used throughout the application above, the argument    zetacutoff

specifies starting values for the ordered beta cutoff parameters; here, the lower cutoffs for each species were set to begin at zero, and the species-common higher cutoff was set to 20. Also, as the ordered beta GLLVM can accommodate exactly 100% records but the kelp forest dataset lacks any observations as such, then we chose to fix the upper cutoff to its starting value:

> setMap  <- list(zeta=c(1:ncol(Yphyl),rep(NA,ncol(Yphyl)))).

This is generally recommended to avoid volatile estimation of the upper cutoff parameter in the absence of exact 100% records in the data. Similar options are available for the dispersion parameters ϕj, using the setting     disp.formula

. In particular, across ‘Structured and correlated community-level row effects’ and ‘Structured and correlated latent variables’ we set    disp.formula=shapeForm

to a vector of ones which leads to a common dispersion parameter across all species of invertebrates, while in ‘Phylogenetic random effect model’ we instead used

> shapeForm  <- ifelse(Traits$GROUP=="INVERT",1,2)

to demonstrate how to share one shared dispersion parameter for invertebrates, and another for seaweed species.

Discussion

In this paper, we presented the newly overhauled gllvm R-package that can be used to address many contemporary questions in modeling community ecology data in a computationally efficient manner. The package has been greatly updated to include a plethora of tools for appropriately modeling and visualizing typical community ecological datasets, a much wider range of distributions for common data types (see also Table A1 in Section ‘Response distributions available in gllvm’ in Appendix), and the capacity to accommodate complex study designs through (for instance) unconstrained ordination at the group-level, incorporating one or more random intercepts to account for pseudo-replication, and by specifying correlation functions across the sampling units due to potential spatial and/or temporal structures. The suite of model-based ordination methods in gllvm has also been extended substantially, including (among others) constrained and concurrent ordination methods that can readily be combined with the community-level effects, ordinations with the aforementioned capabilities to handle nested/hierarchical study designs, and with species-specific effects considered as fixed or (phylogenetically structured) random effects. As demonstrated in this paper, one way in which the last functionality can be useful is when effects need to be “taken out” of the ordination, i.e., the ordination needs to be conditioned on some variable(s). The package also includes other functionalities, e.g., the quadratic response model of Van der Veen et al. (2021) for ordination, which are not covered in this article; these are provided in various vignettes accompanying the package.

gllvm is in continuous development, and there remain many many avenues for further expanding the presented framework and the hence the package. As noted for example in Hui et al. (2023), there remain outstanding challenges in fitting JSDMs to large datasets with many correlated effects (e.g., simultaneous phylogenetic structured species effects and spatio-temporal latent variables, say). Also, the phylogenetic GLLVMs supported by the current version of the package assumes that traits evolve following a Brownian motion. In the future, these models could be extended to support other models for trait evolution, such as the Ornstein–Uhlenbeck Gaussian process model (Uhlenbeck & Ornstein, 1930). Moreover, such models currently rely on a nearest neighbor approximation for computationally scalable estimation, and the authors are currently in the process of adapting the same approximation to scalably fit spatio-temporal GLLVMs (see Tikhonov et al., 2020a for related work using Bayesian estimation). Yet another straightforward but important extension is to handle mixed responses i.e., where the response distributions are allowed to be of mixed type (e.g., Huber, Ronchetti & Victoria-Feser, 2004). Finally, although the gllvm package is able to fit models in parallel, the additional option of being able to utilize GPU resources would further enhane its computational efficiency (see also, Pichler & Hartig, 2021; Rahman et al., 2024).

While this paper’s focus is not a comparison with other packages out there capable of fitting GLLVMs, at this point we do want to acknowledge the increasingly popular glmmTMB R-package (Brooks et al., 2017), which can fit certain types of GLLVMs, although to the best of our knowledge models involving correlated latent variables, correlated random canonical coefficients, phylogenetically structured random slope effects, or quadratic and concurrent GLLVMs have yet to be incorporated into glmmTMB at the time of writing. In the future, gllvm could also learn from glmmTMVB and be extended to include a more general interface to handle zero-inflated and hurdle GLLVMs, although implementation challenges arise if we want to, say, make the latent variables between the zero- and count-component of the model the same or correlated, in order to incorporate correlation between the two model components. Finally, future research could expand the suite of model-based ordination methods available, e.g., by hierarchically modeling the loadings in an ordination (O’Hara & van der Veen, 2024) or by clustering the loadings using a Dirichlet process (e.g., Taylor-Rodríguez et al., 2017; Bystrova et al., 2021). However, likelihood-based estimation of such GLLVMs is (even) more challenging in such extended models, and it may be difficult to find the optimum of the (approximated) likelihood surface due to issues with multimodality and potentially overfitting. Along these lines, gllvm may need to be modified to allow/encourage the use of various penalties to stabilize the fitting process e.g., analogous to the use of weakly informative priors in the Bayesian setting or regularizing penalties to handle complete separation in binary responses (Lemoine, 2019; Clark et al., 2023).

Supplemental Information

Supplemental Information 1 Examples based on the kelp forest data

Supplemental Information 2 Examples based on the ground beetle dataset

Appendix

Response distributions available in gllvm

Table A1 displays the response distributions presently available in gllvm, together with estimation methods (VA, LA, or EVA) and link functions available. Mean–variance relationships associated with each of the models are also included.

Further details about likelihood-based estimation in gllvm

Denote shortly with Ψ and u the vectors collecting all parameters and latent variables in the model, respectively. Considering the core GLLVM Eq. (1) (for simplicity, without random αi’s or βj’s), by assuming that conditional on latent variables ui the responses yij are distributed independently, the complete likelihood function takes the form LΨ;u= ∏i=1n∏j=1mfyij|ui,Ψfui=fy|u,Ψfu,

where f(yij|ui, Ψ) is the conditional distribution of yij and f(ui) are densities of the latent variables. The complete log-likelihood function is then given by logLΨ;u= logfy|u,Ψ+logfu= ∑i=1n ∑j=1m logfyij|ui,Ψ+∑i=1n logfui.

As the complete log-likelihood depends on the unobserved latent variables u, it cannot be maximized using the standard maximum likelihood estimation methods. In gllvm we maximize the marginal likelihood function which is obtained by integrating over the latent variables. As the integral possesses a closed-form only in special cases such as when responses are normally distributed and link function g(⋅) is set to the identity link. As an answer to this, a variety of approximation approaches have been proposed in statistical literature. For a recent review on these, see Korhonen, Nordhausen & Taskinen (2024).

Table A1 Available response distributions in gllvm with the mean and mean–variance functions, E(yij) and V(μij), respectively, estimation methods and link functions for various response types.

	Response	Distribution	Method	Link	Description	
	Binary	Bernoulli	EVA/VA/LA	probit	E(yij) = μij, V(μij) = μij(1 − μij)	
			EVA/LA	logit		
	Non-negative ≥0	Tweedie	LA/(E)VA	log	E(yij) = μij, Vμij=ϕjμijν,	
					where 1 < ν < 2 is a power parameter	
					and ϕj > 0 is a dispersion parameters	
	Counts	Poisson	VA/LA	log	E(yij) = μij, V(μij) = μij	
		NB	VA/LA	log	E(yij) = μij, Vμij=μij+ϕjμij2,	
					where ϕj > 0 is a dispersion parameter	
		ZIP	VA/LA	log	E(yij) = (1 − pj)μij, P(yij = 0) = pj,	
					V(μij) = μij(1 − pj)(1 + μijpj)	
		ZINB	VA/LA	log	E(yij) = (1 − pj)μij, P(yij = 0) = pj,	
					V(μij) = μij(1 − pj)(1 + μij(pj + ϕj))	
		binomial	VA/LA	probit	E(yij) = Njμij, V(μij) = Njμij(1 − μij)	
		binomial	LA	logit	
	Normal	Gaussian	VA/LA	identity	E(yij) = μij, Vyij=ϕj2	
	Ordinal	multinomial	VA	probit	Cumulative probit model	
	Percent cover (0,1)	beta	LA/EVA	probit/logit	E(yij) = μij, V(μij) = μij(1 − μij)/(1 + ϕj)	
	Percent cover [0,1]	ordered beta	EVA	logit	details in Korhonen et al. (2024)	
		Hurdle beta	EVA	logit		
	Positive continuous	gamma	VA/LA	log	E(yij) = μij, Vyij=μij2/ϕj,	
					where ϕj is a shape parameter	
	Positive continuous	exponential	VA/LA	log	E(yij) = μij, Vyij=μij2	

Extended variational approximations

As discussed in Section ‘Estimation’, since the original software article Niku et al. (2019b), gllvm has seen introduction of a new estimation algorithm in the form of extended variational approximations (EVA) as proposed in Korhonen et al. (2023). Here, we will present a slightly more detailed synopsis on EVA, compared to the main text. Of course, for further work, we will direct the reader towards the original article.

To understand how the EVA approach works, we must first briefly review the standard use of VA, as pertaining to GLLVMs. Please refer to likes of Hui et al. (2017), Niku et al. (2019a) and Korhonen, Nordhausen & Taskinen (2024), for further reading. This starts by looking at the evidence lower bound, ℓ(Ψ, ξ), which in likelihood-based estimation of GLLVMs is applied to the marginal log-likelihood function: (7) logLΨ= log∫Rndfy|u,Ψfudu≥Eu∼qlogfy,u|Ψqu|ξ=:ℓΨ,ξ,

derived using Jensen’s inequality (e.g., Needham, 1993). 𝔼u∼q denotes the expected value of u w.r.t. some parameterized variational density q(⋅|ξ). The most common choice for this density—both in general and for GLLVMs specifically—is to take q(⋅|ξ) as a product of multivariate Gaussians with unstructured covariances, but other options also exist in gllvm, namely the matrix normal distribution for the phylogenetic random effects model (Van der Veen & O’Hara, 2024), or the sparse/low-rank variational covariances for models with correlated LVs —see the documentation regarding     gllvm()

and the argument    Lambda.struct

, or https://jenniniku.github.io/gllvm/articles/vignette9.html for a vignette on correlated LVs.

After the parametric form of the variational density is chosen, VA proceeds by maximizing the lower bound ℓ(Ψ, ξ) as a proxy for logLΨ, w.r.t. the parameters of the model and the variational distribution in Ψ and ξ, respectively. Maximizing the lower bound in place of the actual marginal log-likelihood is sensible, as generally, the latter does not possess a tractable form, due to the integral over unobservable quantities. Regretfully, even the lower bound ℓ(Ψ, ξ) very rarely has a closed-form expression, so that it is commonly combined with another form of numerical integration. The culprit here is the first expectation on the right hand size of: (8) Eu∼qlogfy,u|Ψqu|ξ=Eu∼qlogfy|Ψ,u−Eu∼qlogqu|ξfu,

which leads into an integral that needs to be approximated. For the choice of a Gaussian variational density q(⋅|ξ), the latter term corresponds to the negative Kullback–Leibler divergence between two Gaussian distributions—a well-known quantity in statistics. Thus, the EVA approach relies on approximating f(y|Ψ, u), in a way that allows closed-form solution to the expectation for any response type or link function. This is achieved by applying second-order Taylor expansion, around the variational mean, i.e.: (9) logfy|Ψ,u≈ logfy|Ψ,a+∇ulogfy|Ψ,uu=au−a+12u−a⊤∇ u2logfy|Ψ,u u=au−a,

where a is the mean of the variational (Gaussian) distribution q⋅|ξ=N⋅|a,A. Using Eq. (9) in place of the quantity logf(y|Ψ, u) in Eq. (8) yields closed-form approximations to the lower bound, regardless of the response distribution or link (Korhonen et al., 2023). On the other hand, the resulting objective function is no longer guaranteed to be a lower bound to the marginal log-likelihood function, as in Eq. (7) with the regular VA, affecting guarantees of convergence.

Further details on worked examples

Here we provide some additional details relating to some of the examples in Section ‘Worked examples’ of the main text.

Species correlations due to random covariate effects

As an aside note on Section ‘Reduced-rank regression and constrained ordination’ to Section ‘Partial concurrent or constrained ordination’, with (partial) constrained or concurrent ordination GLLVMs, the function    getEnvironCor()

can be used to visualize the species associations that are due to environmental random effects. An example of this, presented in Fig. A1, is produced with the following code:

Figure A1 Plot of species correlations due to 15 random covariate effects.

This plot is obtained from fitting a zero-inflated negative binomial GLLVM with two (fixed) full-rank and 15 random covariate effects constrained on d = 2 latent variables to the ground beetle dataset. The species are ordered according to angular order of the eigenvectors, using setting order=“AOE” for corrplot().

> corrplot::corrplot(getEnvironCor(ftPartOrd), type="lower", diag=FALSE,                              order="AOE", tl.cex=0.7, tl.srt=45, tl.col="black")

where     ftPartOrd

is the partial ordination model fitted to the ground beetle data of Ribera et al. (2001), using the code in Section ‘Partial concurrent or constrained ordination’. Note that here, we get the following warning message when calling    getEnvironCor()

, hinting that a(partial) constrained ordination model Eq. (4) with random effects could be more suitable for our data, rather than one based on concurrent ordination Eq. (5):

Warning  message: In  getResidualCov.gllvm(object, ...) :    The  residual  variance  of CLV1 is very  small. This  might  indicate  that  the  latent  variable  is  nearly  perfectly  represented  by  covariates  alone.

Fitting fourth-corner models in gllvm

To fit a fourth-corner model Eq. (2) on the kelp forest community data of Reed & Miller (2023), with species-specific random effects bj for the two environmental covariates, we need to supply    gllvm()

with the traits via the argument    TR

, and define the environment-trait interactions, i.e., the structure of Bet in Eq. (2) via the    formula

interface:

> ftFourthC  <- gllvm(y=Ykelp,X=Xenv, TR=Traits, family="orderedBeta",                         formula=~logKELP_FRONDSsc + PERCENT_ROCKYsc                         + (logKELP_FRONDSsc + PERCENT_ROCKYsc): (GROUP),                         randomX=~logKELP_FRONDSsc + PERCENT_ROCKYsc,                         num.lv=0, method="EVA", disp.formula=shapeForm,                         starting.val="zero", link="logit", setMap=setMap,                         zetacutoff=c(0,20),  optimizer="nlminb", beta0com=TRUE).

Note the series of terms in     formula

in the code snipped above corresponds exactly to the regression model on the species-specific coefficients in Eq. (2): the first two terms correspond to the effects of the two covariates for the macroalgae group, the next line includes the fourth-corner interaction terms for the invertebrate group, and    randomX

specifies the inclusion of the a species-specific random effects bj. Here, we observe that the estimation of this particular model benefits from using alternative optimizer; that is,    nlminb()

in place of the default    optim()

. Lastly, species-common intercepts are specified with the option    beta0com

.

Phylogenetic tree and covariance matrix

In the example in Section ‘Phylogenetic random effect model’, we estimated fitted a phylogenetic random effect model on the kelp forest community data of Reed & Miller (2023). Estimation of such a model in gllvm requires the user to specify the phylogenetic tree together with the associated covariance matrix, and the matrix of pairwise distances. Additionally, the user may specify the neighbour count used in the nearest Gaussian process approximation (e.g., Datta et al., 2016) that the phylogenetic model relies on. The details on how to setup the aforementioned objects based on the data were omitted from the main text, and are instead shown here. For this, we need the ape R-package:

> phyl  <- ape::as.phylo(~Kingdom/Phylum/Class/Order/Family/Genus/Species, data=kelptaxa) > tree  <- ape::compute.brlen(phyl)      # the  phylogenetic  tree > colMat  <- ape::vcv(tree)                # the  covariance  matrix > dist  <- ape::cophenetic.phylo(tree)   # pairwise  distances.

The     colMat

now contains essentially the matrix C seen in Eq. (6), and needs to be supplied to    gllvm()

in order to fit the model, same as    dist

, which contains the pairwise distances between the tips of our phylogenetic tree. Based on the tree, we the need to define some ordering for the species, for example with:

> allDists  <- ape::dist.nodes(tree) > order  <- as.integer(names(sort(allDists[1:length(tree$tip.label),             nrow(allDists)],  decreasing=TRUE))) > order  <- tree$tip.label[order]

to order species according to their distance to the root of the tree.

Because such models tend to be computationally intensive to fit, and thus slow, the gllvm R-package makes use of an additional approximation. As a downside to this approximation, we have to determine the number of neighbouring species to consider for each species, which can be set using the argument     nn.colMat

argument in the call to    gllvm()

. A higher number of neighbors is more accurate, but slower; around 10–15 neighbors is usually a good and balanced choice (Van der Veen & O’Hara, 2024), and the software defaults to 10. Note, that the quality of this approximation is sensitive to the ordering of the species in the data, which can have a considerable effect on the results. This is covered in more detail in the associated vignette on the phylogenetic model at https://jenniniku.github.io/gllvm/articles/vignette7.html.

Additional Information and Declarations

Competing Interests

Author Contributions

Data Availability

The authors declare there are no competing interests.

Pekka Korhonen conceived and designed the experiments, performed the experiments, analyzed the data, prepared figures and/or tables, authored or reviewed drafts of the article, and approved the final draft.

Francis K.C. Hui conceived and designed the experiments, prepared figures and/or tables, authored or reviewed drafts of the article, and approved the final draft.

Jenni Niku conceived and designed the experiments, authored or reviewed drafts of the article, and approved the final draft.

Sara Taskinen conceived and designed the experiments, prepared figures and/or tables, authored or reviewed drafts of the article, and approved the final draft.

Bert van der Veen conceived and designed the experiments, performed the experiments, prepared figures and/or tables, authored or reviewed drafts of the article, and approved the final draft.

The following information was supplied regarding data availability:

The code for the examples in ‘Worked Examples’ are available in the Supplementary Files. The data used in the examples are available with the package.

Original references for these datasets are also included in the manuscript.

The ground beetle data is found in Ecological Archives E082-012: https://esapubs.org/archive/ecol/E082/012/.

The kelp forest dataset can be accessed from https://doi.org/10.6073/pasta/0af1a5b0d9dde5b4e5915c0012ccf99c.

Both of these are also part of the ‘gllvm’ R package featured in the manuscript, which itself is available from CRAN https://doi.org/10.32614/CRAN.package.gllvm.

The ’gllvm’ package is available at CRAN, GitHub and Zenodo:

- https://github.com/JenniNiku/gllvm.

- Niku, J., van der Veen, B., Hui, F. K. C., Korhonen, P., Taskinen, S., Brooks, W., & Warton, D. I. (2025). gllvm: Generalized Linear Latent Variable Models. R package (V2.0.4). Zenodo. https://doi.org/10.5281/zenodo.15720641

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
