# Peer review of "gllvm 2.0: fast fitting of advanced ordination methods and joint species distribution models"

_PeerJ, doi:10.7717/peerj.20338_

## Round 0.1 · original submission · Major Revisions

· Academic Editor

Major Revisions

Dear Dr. Korhonen, I ask you to follow the program very carefully and clarify those parameters that do not allow the reviewer to obtain the same results as you. The readers of the article will have the same problems as the reviewer. I hope that you will make the necessary changes to this article.

·

Basic reporting

In their article, the authors consider multidimensional methods based on models that extend generalised linear models to multiple response variables, taking into account the correlation between responses. Particular attention is paid to the application of Generalised Linear Latent Variable Models in community ecology. It is certainly worth agreeing with the authors that the R package gllvm can now be considered a general framework for joint modelling of community ecology data.
The literature review is comprehensive and informative, providing an overview of the current state of the problem, unresolved issues, and the authors' proposed solutions. The structure of the article fully complies with PeerJ standards.

Lines 43–44: A sentence ‘The data are often very sparse, because species often occur at few places due to, for example, environmental filtering’ should have a logical continuation, otherwise it looks like something taken out of context. It is possible that the paragraph transition (Line 45) is unnecessary, as a new paragraph does not usually begin with words such as ‘Instead’.

Lines 50–51 ‘Classically, community ecological studies visualise a small number of latent variables to describe patterns of species co-occurrence as well as the (dis)similarity of sites, using ordination’ – It may be worth specifying that this refers to indirect ordination, because direct ordination uses manifest (actually observable) variables rather than latent ones. It also makes sense to clarify, if the authors deem it necessary, that latent variables are proxy estimates of environmental factors: in mathematical terms, we are talking about latent variables, while in ecological terms, we are talking about environmental gradients.

Line 60: ‘GLLVMs have been consistently shown to outperform traditional unconstrained ordination techniques …’ – It would be interesting to clarify what exactly is the advantage of this approach compared to other methods of ordination?

Experimental design

The authors consider several working examples as cases for applying the package. They are interesting and provide an idea of the package's capabilities and features.

Validity of the findings

With the development of the gllvm package, new features have emerged that fundamentally expand its capabilities and necessitate the justification and explanation of such innovations, which is done in this article. The authors have conducted a comparative analysis of the new features of the package. It is clear that the innovations have significantly improved the package's ability to solve problems with obvious environmental specificity. The authors note that fitting GLLVMs is generally a computationally burdensome task. Undoubtedly, this feature significantly limited the application of GLLVMs, since, in addition to purely computational limitations, there were limitations to the application of this approach related to the mathematical competence of ecologists. The authors offer their vision for solving this problem, which is very important for the dissemination of the new approach among a wide range of researchers.

Reviewer 2 ·

Basic reporting

The manuscript is well motivated and clearly written. Some points:

1. I cannot see that it is stated anywhere that the gllvm package is available from the Comprehensive R Archive Network (CRAN). The reader deserves to know this.
2. Please cite R itself. See the result of invoking the command "citation()" inside R.
3. In the abstract, "Background" section, change "a powerful tools" to "powerful tools".
4. Line 36: Your citation makes it look as if Skrondal and Rabe-Hesketh used the term GLLVM, which they didn't. They used the term GLLAMM.
5. Line 137: Change "Makov" to "Markov".
6. Line 138: You should probably have introduced the abbreviation MCMC on the line above.
7. Line 159: Rewrite "shown good to perform well" to "shown to perform well".
8. Line 162: Rewrite "under variational framework" to "under the variational framework" or "under a variational framework".
9. Line 332: Change "these can also shared" to "these can also be shared".
10. Line 342: Change "also varying the varying number" to "also varying the number".
11. Line 426: Change "related" to "relates".
12. Line 461: Maybe a matter of taste, but you might want to leave it to the reader/user to decide whether the approximations are "clever".
13. Line 621: Change "gllvm may be need to be modified" to "gllvm may need to be modified".

Experimental design

no comment

Validity of the findings

Here I have major concerns. Of note, I am running on R 4.5.0 and using version 2.0.2 installed from CRAN.

1. Neither the reproduction scripts nor the manuscript tell us exactly how to install the package. For future readers, it is also necessary to know which version of the package. To ensure reproducibilty also in the future, you should tag a release version on GitHub containing exactly the package as it was on submission of this manuscript, and the instruct the reader how to install this using something likes remotes::install_github(). Since the package is likely to change in the future, simply installing from CRAN is likely to yield different results a few years from now.

2. Running your script for the beetle exampel, with exactly the same random number seed, I only reproduce the lower left plot in Figure 1. The three other plots look completely different.

3. I get a WARNING on line 35 of the beetle reproduction script. It looks to me as it both suggests a serious error (message 1) and that the model is empirically underidentified (message 2):

> ftPartOrd <- gllvm(y=beetle, X=X, family="ZINB", num.lv.c=2,
+ formula=~Canopyheight + Reprobiom,
+ lv.formula=~Texture + Org + pH + AvailP + AvailK + Moist
+ + Bare + Litter + Bryophyte + Plants.m2 + Stemdensity + Biom_l5
+ + Biom_m5 + Elevation + Management, randomB="P", n.init=5)
Warning messages:
1: In attr(term, "term.labels") == labterm :
longer object length is not a multiple of shorter object length
2: In doTryCatch(return(expr), name, parentenv, handler) :
Determinant of the variance-covariance matix is zero. Please double check your model for e.g. overfitting or lack of convergence.

4. I do not get the same numbers as in lines 293-309 from the reproduction script, even when using your random number seed.

5. I cannot reproduce Figure 5 using your script.

6. I get the following error at the end of the beetle reproduction script:

> # model selection using information criteria:
> lvcount <- c(1,2,3,4)
> model <- c("poisson", "ZIP", "negative.binomial", "ZINB")
> res <- data.frame(family = rep(model, each=4), num.lv.c = rep(lvcount, 4),
+ AIC = NA, AICc = NA, BIC = NA, logL = NA, df = NA)
> idx = 1
> for (m in model) {
+ for (d in lvcount) {
+ ft <- gllvm(y = beetle, X = X, family = m, num.lv.c = d)
+ res$AIC[idx] <- AIC(ft)
+ res$AICc[idx] <- AICc(ft)
+ res$BIC[idx] <- BIC(ft)
+ res$logL[idx] <- ft$logL
+ res$df[idx] <- summary(ft)$df
+ idx <- idx+1
+ }
+ }
Error in FAstart(eta = mu, family = family, y = y, num.lv = num.lv, num.lv.c = num.lv.c, :
Calculation of starting values has failed.
In addition: Warning message:
In gllvm(y = beetle, X = X, family = m, num.lv.c = d) :
Selected optimizer not available for this model. Using optim instead.


7. In the kelp forest reproduction script I start by getting the following warning:

> ftStrucRow <- gllvm(y = Ysess, X = Xenv, family = "orderedBeta",
+ formula = ~ logKELP_FRONDSsc + PERCENT_ROCKYsc,
+ studyDesign = Xenv[,c("SITE", "TRANSECT", "YEAR")],
+ row.eff = ~ YEAR + (1|SITE/TRANSECT),
+ num.lv = 0, method = "EVA", link = "logit",
+ disp.formula = shapeForm, setMap = setMap,
+ control.start = list(zetacutoff = c(0, 20)))
Warning messages:
1: In nlminb(objr$par, objr$fn, objr$gr, control = list(rel.tol = reltol, :
NA/NaN function evaluation
2: In nlminb(objr$par, objr$fn, objr$gr, control = list(rel.tol = reltol, :
NA/NaN function evaluation

8. Using the reproduction script with the seed that you set, I can neither reproduce Figure 3. For example, LV2 mean equals 22.7% in my run, which is not a minor difference.

9. I also get warnings with the fourth corner model:

> ftFourthC <- gllvm(y = Ykelp, X = Xenv, TR = Traits,
+ formula = ~ logKELP_FRONDSsc + PERCENT_ROCKYsc +
+ (logKELP_FRONDSsc + PERCENT_ROCKYsc) : (GROUP) +
+ (0+logKELP_FRONDSsc + PERCENT_ROCKYsc|1), num.lv = 0,
+ disp.formula = shapeForm, starting.val = "zero",
+ family = "orderedBeta", link = "logit", method = "EVA",
+ zetacutoff = c(0,20), setMap = setMap)
Warning messages:
1: In data.frame(..., check.names = FALSE) :
row names were found from a short variable and have been discarded
2: In data.frame(..., check.names = FALSE) :
row names were found from a short variable and have been discarded

10. I get an error on line 131 of the kelp forest reproduction script, which does not allow me to proceed further:
> shapeForm = ifelse(Trphyl[,order]$GROUP == "INVERT", 1, 2)
Error in `[.data.frame`(Trphyl, , order) : undefined columns selected

Additional comments

The paper is well written, but all the issues related to the reproduction script suggests that the models are underidentified. That is, they are too complex for the given amount of data, and the optimization algorithm is not capable of finding a clear optimum. This must be carefully revised, maybe by using different data or simpler models. You should also make sure that the results reproduce on both Windows and Unix based platforms.

Since the main scientific work described in this paper is the gllvm package, all the examples should be such that the package issues no warnings or errors when the user runs through your examples by stepping through your reproduction script.

·

Basic reporting

This is a well-written paper with just enough background to understand the novel contributions in version 2 of the gllvm package. A full understanding of the models and their interpretation is a huge topic. Leaving that to other sources is appropriate. The figures are useful and mostly high-quality. Some issues are noted below. Raw data are supplied with the gllvm package.

Experimental design

Not relevant. This manuscript describes a major revision of an R package.

Validity of the findings

Not relevant. This manuscript describes a major revision of an R package.

Additional comments

Minor details – all are intended to improve clarity:

Your figures show a lot of information. This is to be expected given the complexity of the patterns that are being illustrated. However, some features are hard to see. For example, the legends describing the biplot panels in Figures 1 and 2 mention “arrows … are shown in darker color”. Because of the overlap, it is nearly impossible to see what is in a darker color, even in the color pdf. I think I see one darker label, Repro???, in Figure 2, but I can’t tell whether there are any others. And, many labels overprint. I don’t know what software you are using to draw the biplots, but the vegan plot functions use code to shift labels to avoid overplotting.

Do the top panels in Figure 1 include all species? I was expecting 68. Clarity is important because your figures and text will be used as templates by others learning to use the package.

lines 293 et seq. The text on line 290 et seq mentions that this table illustrates shrinkage by contrasting these coefficients with those from earlier fixed effect models. I didn’t see the earlier coefficients given anywhere. You could explicitly illustrate shrinkage by adding the earlier coefficients to this table.

line 859-860: Your text describes the logKELP_FRONDsc and PERCENT_ROCKYsc coefficients as “main effects”. This is a frequent mis-interpretation of R coefficients with the default R contr.treatment coding of indicator variables. Those coefficients are the regression coefficients for the Algae group (alphabetically first). Traditionally, main effects are averages over omitted factors.

---

## Round 0.2 · Minor Revisions

· Academic Editor

Minor Revisions

Dear Dr. Korhonen,

Please make technical clarifications to the version of the statistical analysis package as requested by the reviewer.

Reviewer 2 ·

Basic reporting

All good.

Experimental design

All good.

Validity of the findings

The authors have addressed my concerns.

·

Basic reporting

The original manuscript was weel written. The revisions have improved the manuscript clarity.

Experimental design

Not appropriate. This manuscript describes a major revision of an R package.

Validity of the findings

Not appropriate. This manuscript describes a major revision of an R package. The dependence on R version (4.4 used in the manuscript) and different results for reviewer 2 (version 4.5) are concerning. You indicate the version you used but not the fact that different versions give different results. At a minimum, you should rndicate this can happen and recommend that reporting version numbers is crucia. It is good practice but not always done.l. My guess is that the optimizer code changed between R versions, which you can not control

Additional comments

Minor details:
Figure 1 legend, top left panel. Legend refers to “darker color”, which was appropriate for the earlier version. Arrows are now in red, which is much easier to see.

---

## Round 0.3 · accepted · Accept

· Academic Editor

Accept

Dear Dr. Korhonen, I congratulate you on the acceptance of this article for publication.